# LowKey: Leveraging Adversarial Attacks to Protect Social Media Users from Facial Recognition

**Valeriia Cherepanova**
Department of Mathematics
University of Maryland
vcherepa@umd.edu

**Micah Goldblum**
Department of Computer Science
University of Maryland
goldblum@umd.edu

**Harrison Foley**[*]
Department of Computer Science
US Naval Academy
m211926@usna.edu

**Shiyuan Duan**[*]
Department of Computer Science
University of Maryland
sduan1@umd.edu

**John Dickerson**
Department of Computer Science
University of Maryland
john@cs.umd.edu

**Gavin Taylor**
Department of Computer Science
US Naval Academy
taylor@usna.edu

**Tom Goldstein**
Department of Computer Science
University of Maryland
tomg@cs.umd.edu

## Abstract

Facial recognition systems are increasingly deployed by private corporations, government agencies, and contractors for consumer services and mass surveillance programs alike. These systems are typically built by scraping social media profiles for user images. Adversarial perturbations have been proposed for bypassing facial recognition systems. However, existing methods fail on full-scale systems and commercial APIs. We develop our own adversarial filter that accounts for the entire image processing pipeline and is demonstrably effective against industrial-grade pipelines that include face detection and large scale databases. Additionally, we release an easy-to-use webtool that significantly degrades the accuracy of Amazon Rekognition and the Microsoft Azure Face Recognition API, reducing the accuracy of each to below 1%.

## 1 Introduction

Facial recognition systems (FR) are widely deployed for mass surveillance by government agencies, government contractors, and private companies alike on massive databases of images belonging to private individuals (Hartzog, 2020; Derringer, 2019; Weise & Singer, 2020). Recently, these systems have been thrust into the limelight in the midst of outrage over invasion into personal life and concerns regarding fairness (Singer, 2018; Lohr, 2018; Cherepanova et al., 2021). Practitioners populate their databases by hoarding publicly available images from social media outlets, and so users are forced to choose between keeping their images outside of public view or taking their chances with mass surveillance.

We develop a tool, LowKey, for protecting users from unauthorized surveillance by leveraging methods from the adversarial attack literature, and make it available to the public as a webtool.

---

[*]Authors contributed equally.

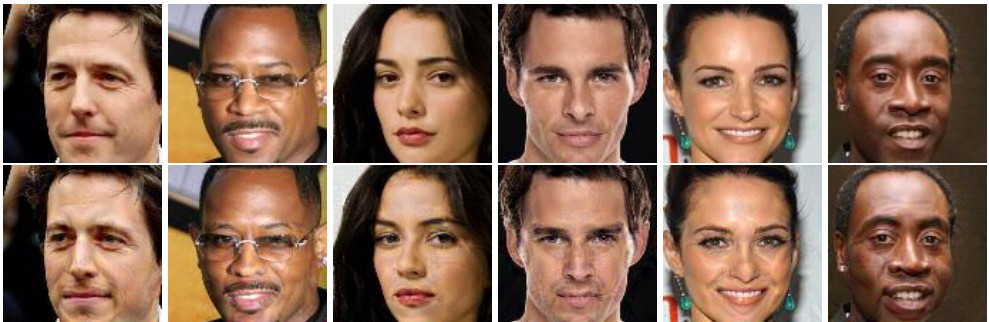

Figure 1: Top: original images, Bottom: protected by LowKey.

LowKey is the first such evasion tool that is effective against commercial facial recognition APIs. Our system pre-processes user images before they are made publicly available on social media outlets so they cannot be used by a third party for facial recognition purposes. We establish the effectiveness of LowKey throughout this work.

Our contributions can be summarized as follows:

- We design a black-box adversarial attack on facial recognition models. Our algorithm moves the feature space representations of gallery faces so that they do not match corresponding probe images while preserving image quality.
- We interrogate the performance of our method on commercial black-box APIs, including Amazon Rekognition and Microsoft Azure Face, whose inner workings are not publicly known. We provide comprehensive comparisons with the existing data poisoning alternative, Fawkes (Shan et al., 2020), and we find that while Fawkes is ineffective in every experiment, our method consistently prevents facial recognition.
- We release an easy-to-use webtool, LowKey, so that social media users are no longer confronted with a choice between withdrawing their social media presence from public view and risking the repercussions of being surveilled.

## 2 RELATED WORK

Neural networks are known to be vulnerable to *adversarial attacks*, small perturbations to inputs that do not change semantic content, and yet cause the network to misbehave (Goodfellow et al., 2014). The adversarial attack literature has largely focused on developing new algorithms that, in simulations, are able to fool neural networks (Carlini & Wagner, 2017; Chiang et al., 2020). Most works to date focus on the idea of *physical world attacks*, in which the attacker places adversarial patterns on an object in hopes that the adversarial properties transfer to an image of the object. Such attacks do not succeed reliably because the adversarial perturbation must survive imaging under various lighting conditions, object orientations, and occlusions (Kurakin et al., 2016). While researchers have succeeded in crafting such attacks against realistic systems, these attacks do not work consistently across environments (Wu et al., 2019; Xu et al., 2019; Goldblum et al., 2020). In facial recognition, attacks have largely focused on physical backdoor threat models, evasion attacks on verification (Wenger et al., 2020; Zhong & Deng, 2020) and attacks on face detection (Pedraza et al., 2018). Unlike these physical threat models, the setting in which we operate is purely *digital*, meaning that we can manipulate the contents of digital media at the bit level, and then hand manipulated data directly to a machine learning system. The ability to digitally manipulate media greatly simplifies the task of attacking a system, and has been shown to enhance transferability to black box industrial systems for applications like copyright detection (Saadatpanah et al., 2020) and financial time series analysis (Goldblum et al., 2020).

Recently, the Fawkes algorithm was developed for preventing social media images from being used by unauthorized facial recognition systems (Shan et al., 2020). However, Fawkes, along with the experimental setup on which it is evaluated in the original work, suffers from critical problems. First, Fawkes assumes that facial recognition practitioners train their models on each individual's data.

However, high-performance FR systems instead harness large pre-trained Siamese networks (Liu et al., 2017; Deng et al., 2019). Second, the authors primarily use image *classifiers*. In contrast, commercial systems are trained with FR-specific heads and loss functions, as opposed to the standard cross-entropy loss used by classifiers. Third, the authors perform evaluations on very small datasets. Specifically, they test Fawkes against commercial APIs with a gallery containing only 50 images. Fourth, the system was only evaluated using top-1 accuracy, but FR users such as police departments often compile a list of suspects rather than a single individual. As a result, other metrics like top-50 accuracy are often used in facial recognition, and are a more realistic metric for when a system has been successfully suppressed. Fifth, while the original work portrays Fawkes' perturbations are undetectable by the human eye, experience with the codebase suggests the opposite (indeed, a New York Times journalist likewise noted that the Fawkes images she was shown during a demonstration were visibly heavily distorted). Finally, Fawkes has not yet released an app or a webtool, and regular social media users are unlikely to make use of git repositories. Our attack avoids the aforementioned limitations, and we perform thorough evaluations on a large collection of images and identities. When comparing with Fawkes, we use the authors' own implementation in order to make sure that all evaluations are fair. Furthermore, we use Fawkes' highest protection setting to make sure that LowKey performs better than Fawkes' best attack. Another work uses targeted adversarial attack on probe images for facial recognition systems so that they cannot be matched with images in a database (Yang et al., 2020).

## 3 THE LOWKEY ATTACK ON MASS SURVEILLANCE

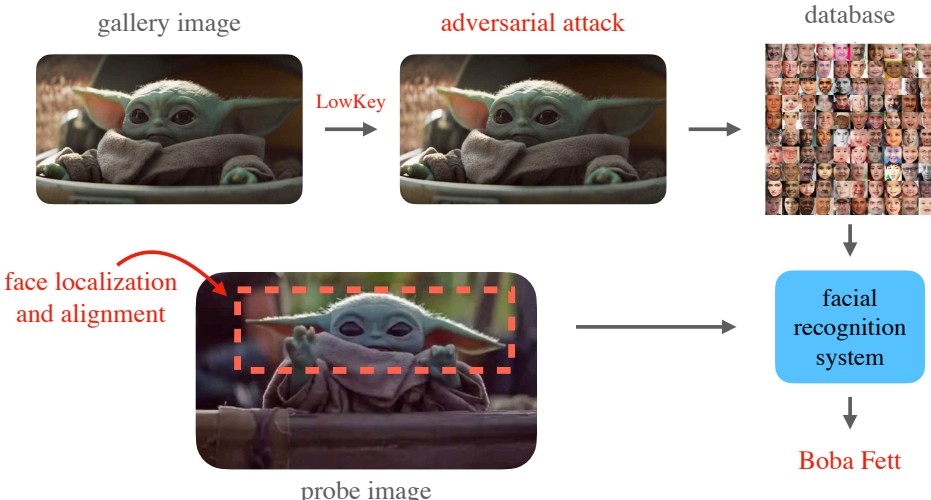

Figure 2: The LowKey pipeline. When users protect their publicly available images with LowKey, facial recognition systems cannot match these harvested images with new images of the user, for example from surveillance cameras.

### 3.1 PROBLEM SETUP

To help make our work more widely accessible, we begin by introducing common facial recognition terms.

***Gallery images*** are database images with known identities. These often originate from such sources as passport photos and social media profiles. The gallery is used as a reference for comparing new images.

***Probe images*** are new photos whose subject the FR system user wants to identify. For example, probe images may be extracted from video surveillance footage. The extracted images are then fed into the FR system, and matches to gallery images with known identities.

***Identification*** is the task of answering the question, "who is this person?" Identification entails comparing a probe image to gallery images in order to find potential matches. In contrast, **verification** answers the question, "is this person who they say they are?", or equivalently "are these two photos of the same person?" Verification is used, for example, to unlock phones.

In our work, we focus on identification, which can be used for mass surveillance. State-of-the-art facial recognition systems first detect and align faces before extracting facial features from the probe image using a neural network. These systems then find gallery images with the closest feature vectors using a $k$-nearest neighbors search. The matched gallery images are then considered as likely identities corresponding to the person in the probe photo. LowKey applies a filter to user images which may end up in an organization's database of gallery images. The result is to corrupt the gallery feature vectors so that they will not match feature vectors corresponding to the user's probe images. A visual depiction of the LowKey pipeline can be found in Figure 2.

### 3.2   THE LOWKEY ATTACK

LowKey manipulates potential gallery images so that they do not match probe images of the same person. LowKey does this by generating a perturbed image whose feature vector lies far away from the original image, while simultaneously minimizing a perceptual similarity loss between the original and perturbed image. Maximizing the distance in feature space prevents the image from matching other images of the individual, while the perceptual similarity loss prevents the image quality from degrading. In this section, we formulate the optimization problem, and describe a number of important details.

LowKey is designed to evade proprietary FR systems that contain pre-processing steps and neural network backbones that are not publicly known. In order to improve the transferability of our attack to unknown facial recognition systems, LowKey simultaneously attacks an ensemble of models with various backbone architectures that are produced using different training algorithms. Additionally, for each model in the ensemble, the objective function considers the locations of feature vectors of the attacked image both with and without a Gaussian blur. We find that this technique improves both the appearance and transferability of attacked images. Experiments and ablations concerning ensembling and Gaussian smoothing can be found in Section 6. For perceptual similarity loss, we use LPIPS, a metric based on $\ell_2$ distance in the feature space of an ImageNet-trained feature extractor (Zhang et al., 2018). LPIPS has been used effectively in the image classification setting to improve the image quality of adversarial examples (Laidlaw et al., 2020).

Formally, the optimization problem we solve is

$$\max_{x'} \frac{1}{2n} \sum_{i=1}^{n} \frac{\overbrace{\|f_i(A(x)) - f_i(A(x'))\|_2^2}^{\text{non-smoothed}} + \overbrace{\|f_i(A(x)) - f_i(A(G(x')))\|_2^2}^{\text{smoothed}}}{\|f_i(A(x))\|_2} - \alpha \underbrace{\text{LPIPS}(x, x')}_{\text{perceptual loss}}, \quad (1)$$

where $x$ is the original image, $x'$ is the perturbed image, $f_i$ denotes the $i^{th}$ model in our ensemble, $G$ is the Gaussian smoothing function with fixed parameters, and $A$ denotes face detection and extraction followed by $112 \times 112$ resizing and alignment. The face detection step is an important part of the LowKey objective function, as commercial systems rely on face detection and extraction because probe images often contain a scene much larger than a face, or else contain a face who's alignment is not compatible with the face recognition system.

We solve this maximization problem iteratively with signed gradient ascent, which is known to be highly effective for breaking common image classification systems (Madry et al., 2017). Namely, we iteratively update $x'$ by adding the sign of the gradient of the maximization objective (1) with respect to $x'$. By doing this, we move $x'$ and $G(x')$ far away from the original image $x$ in the feature spaces of models $f_i$ used in the LowKey ensemble. The ensemble contains four feature extractors, IR-152 and ResNet-152 backbones trained with ArcFace and CosFace heads. More details can be found in the next section.

Additional details concerning attack hyperparameters can be found in Appendix 8.1.

## 4  EXPERIMENTAL DESIGN

Our ensemble of models contains ArcFace and CosFace facial recognition systems (Deng et al., 2019; Wang et al., 2018). For each of these systems, we train ResNet-50, ResNet-152, IR-50, and IR-152 backbones on the MS-Celeb-1M dataset, which contains over five million images from over 85,000 identities (He et al., 2016; Deng et al., 2019; Guo et al., 2016). We use these models both in our ensemble to generate attacks and to perform controlled experiments in Section 6. Additional details on our models and their training routines can be found in Appendix 8.1.

We primarily test our attacks on the FaceScrub dataset, a standard identification benchmark from the MegaFace challenge, which contains over 100,000 images from 530 known identities as well as one million distractor images (Kemelmacher-Shlizerman et al., 2016). We discard near-duplicate images from the dataset as is common practice in the facial recognition literature (Zhang et al., 2020). We also perform experiments on the UMDFaces dataset, which can be found in Appendix 8.3 (Bansal et al., 2017). We treat one tenth of each identity's images as probe images, and we insert the remaining images into the gallery. We randomly select 100 identities and apply LowKey to each of their gallery images. This setting simulates a small pool of LowKey users among a larger population of non-users. Then, in order to perform a single evaluation trial of identification, we randomly sample one probe image from a known identity and find its closest matches within the remainder of the FaceScrub dataset, according to the facial recognition model. Distance is measured in feature space of the model. If the FR model selects a match from the same identity, then the trial is a success.

**Note 1** (Rank-$k$ Accuracy). *For each probe image, we consider the model successful in the rank-$k$ setting if the correct identity appears among the $k$ closest gallery images in the model's feature space. To test the transferability of our attack we compute rank-1 and rank-50 accuracy for attack, and test feature extractors from our set of trained FR models.*

## 5  BREAKING COMMERCIAL BLACK-BOX APIS

The ultimate test for our protection tool is against commercial systems. These systems are proprietary, and their exact specifications are not publicly available. We test LowKey in the black-box setting using two commercial facial recognition APIs: Amazon Rekognition and Microsoft Azure Face. We also compare against Fawkes. We generate Fawkes images using the authors' own code and hyperparameters to ensure a fair comparison, and we use the highest protection setting their code offers.

**Amazon Rekognition** Amazon Rekognition is a commercial tool for detecting and recognizing faces in photos. Rekognition works by matching probe images with uploaded gallery images that have known labels. Amazon does not describe how their algorithm works, but their approach seemingly does not involve training a model on uploaded images (at least not in a supervised manner). We test the Rekognition API using the FaceScrub dataset (including distractors) where 100 randomly selected identities have their images attacked as described in Section 4. We observe that LowKey is highly effective, and even in the setting of rank-50 accuracy, Rekognition can only recognize 2.4% of probe images belonging to users protected with LowKey. In contrast, Fawkes fails, with 77.5% of probe images belonging to its users recognized correctly in the rank-1 setting and 94.9% of these images recognized correctly when the 50 closest matches are considered. This is close to the performance of Amazon Rekognition on clean images.

| | Amazon rank-1 | Amazon rank-50 | Microsoft rank-1 |
|---|---|---|---|
| Clean | 93.7% | 95.4% | 90.5% |
| Fawkes | 77.5% | 94.9% | 74.2% |
| LowKey | 0.6% | 2.4% | 0.1% |

Table 1: An evaluation of Amazon Rekognition and Microsoft Azure Face on FaceScrub data with LowKey and Fawkes protection (a small number, and lighter color, indicates a successful attack). LowKey consistently achieves virtually flawless protection, while Fawkes provides little protection.

**Microsoft Azure Face** We repeat a similar experiment on the Microsoft Azure Facial Recognition API. In contrast to Amazon's API, Microsoft updates their model on the uploaded gallery of images. Therefore, only known identities can be used, so we only include images corresponding to the 530 known identities from FaceScrub and no distractors. The Azure system recognizes only 0.1% of probe images whose gallery images are under the protection of LowKey. Even though Fawkes is designed to perform data poisoning, and authors claim it is especially well suited to Microsoft Azure Face, in our experiments, Azure is still able to recognize more than 74% of probe images uploaded by users who employ Fawkes.

We conclude from these experiments that LowKey is both highly effective and transferable to even state-of-the-art industrial facial recognition systems. In the next section, we explore several components of our attack in order to uncover the tools of its success.

## 6 ADDITIONAL EXPERIMENTS

The effectiveness of our protection tool hinges on several properties:

1. The attack must transfer effectively to unseen models.
2. Images must look acceptable to users.
3. LowKey must run sufficiently fast so that run-time does not outweigh its protective benefits.
4. Attacked images must remain effective after being saved in PNG and JPG formats.
5. The algorithm must scale to images of any size.

We conduct extensive experiments in this section with a variety of facial recognition systems to interrogate these properties of LowKey.

### 6.1 ENSEMBLES AND TRANSFERABILITY

In developing the ensemble of models used to compute our attack, we examine the extent to which attacks generated by one model are effective against another. By including an eclectic mix of models in our ensemble, we are able to ensure that LowKey produces images that fool a wide variety of facial recognition systems. To this end, we evaluate attacks on all pairs of source and victim models with ResNet-50, ResNet-152, IR-50, and IR-152 backbones, and both ArcFace and CosFace heads. For each victim model, we additionally measure performance on clean images, our ensembled attack, and Fawkes. See Table 2 for a comparison of the rank-50 performance of these combinations. Additional evaluations in the rank-1 setting and on the UMDFaces dataset can be found in Appendix 8.2 and 8.3 respectively. Note that entries for which the attacker and defender models are identical depict white-box performance, while entries for which these model differ depict black-box transferability.

We observe in these experiments that adversarial attacks generated by IR architectures transfer better to IR-based facial recognition systems, while attacks generated by ResNet architectures transfer better to other ResNet systems. In general, attacks computed on 152-layer backbones are more effective than attacks computed on 50-layer backbones, and deeper networks are also more difficult to fool. Moreover, attacks transfer better between models trained with the same head. An ensemble of models of all combinations of ResNet-152 and IR-152 backbones as well as ArcFace and CosFace heads generates attacks that transfer effectively to all models and fool models at only a slightly lower rate than white-box attacks.

### 6.2 GAUSSIAN SMOOTHING

We incorporate Gaussian smoothing as a pre-processing step in our objective function (1) to make our perturbations smoother and more robust. Intuitively, this promotes the effectiveness of the attacked image even when a denoising filter is applied. The presence of blur forces the adversarial perturbation to rely on smoother/low-frequency image modifications rather than adversarial "noise." Empirically, we find that attacks computed with this procedure produce slightly smoother and more aesthetically pleasing perturbations without sharp lines and high-frequency oscillations. See Figure 3 for a visual comparison of images produced with and without Gaussian smoothing in the LowKey pipeline.

|  | Defender | | | | | | | |
|  | IR-50A | IR-50C | IR-152A | IR-152C | RN-50A | RN-50C | RN-152A | RN-152C |
|---|---|---|---|---|---|---|---|---|
| Clean | 96.8% | 96.8% | 96.7% | 96.8% | 96.8% | 96.8% | 96.7% | 96.7% |
| Fawkes | 96.6% | 96.7% | 96.7% | 96.7% | 96.7% | 96.5% | 96.6% | 96.6% |
| IR-50A | 0.4% | 22.2% | 11.9% | 35.2% | 33.6% | 46.4% | 45.7% | 53.0% |
| IR-50C | 4.9% | 0.3% | 4.1% | 8.0% | 23.1% | 25.9% | 31.4% | 28.6% |
| IR-152A | 9.9% | 18.3% | 0.1% | 26.1% | 41.6% | 46.2% | 49.6% | 48.9% |
| IR-152C | 2.8% | 1.6% | 1.5% | 0.5% | 11.9% | 13.9% | 18.8% | 16.3% |
| RN-50A | 26.4% | 35.7% | 36.3% | 43.0% | 0.9% | 13.3% | 17.4% | 24.2% |
| RN-50C | 33.8% | 36.5% | 41.1% | 42.9% | 9.9% | 0.2% | 17.9% | 21.1% |
| RN-152A | 16.8% | 22.0% | 21.1% | 28.2% | 5.2% | 8.8% | 0.3% | 7.6% |
| RN-152C | 14.8% | 19.2% | 19.9% | 24.3% | 6.7% | 6.9% | 7.1% | 0.5% |
| Ensemble | 3.0% | 2.4% | 2.1% | 0.6% | 3.1% | 4.2% | 5.5% | 0.9% |

(The leftmost label "Attacker" runs vertically beside the row labels.)

Table 2: Rank-50 accuracy of the LowKey and Fawkes attacks. After the first two rows, each row represents LowKey attacks generated from the same model. Each column represents inference on a single model. The first two letters in the model's name denote the type of backbone: IR or ResNet (RN). The last letter in the model's name indicates the type of head; "A" denotes ArcFace, and "C" denotes CosFace. Smaller numbers, and lighter colors, indicate more successful attacks.

We additionally produce images both with and without smoothing in the attack pipeline. Before feeding them into facial recognition systems, we defend the system against our attacks by applying a Gaussian smoothing pre-processing step just before inference.

We find that facial recognition systems which use this pre-processing step perform equally well on rank-50 (but not rank-1) accuracy compared to performance without smoothing, and they are also able to defeat attacks which are not computed with Gaussian smoothing. On the other hand, attacks computed using Gaussian smoothing are able to counteract this defense and fool the facial recognition system (see Table 3). This suggests that attacks that use Gaussian smoothing in their pipeline are more robust and harder to defend against. See Appendix 8.6 for details regarding Gaussian smoothing hyperparameters.

## 6.3 RUN-TIME

In order for users to be willing to use our tool, LowKey must run fast enough that it is not an inconvenience to use. Computing adversarial attacks is a computationally expensive task. We compare run-time to Fawkes as a baseline and test both attacks on a single NVIDIA GeForce RTX 2080 TI GPU. We attack one image at a time with no batching for fair comparison, and we average over runs on every full-size gallery image from each of five randomly selected identities from FaceScrub. While Fawkes averages 54 seconds per image, LowKey only averages 32 seconds per image. In addition to providing far superior protection, LowKey runs significantly faster than the existing method, providing users a smoother and more convenient experience.

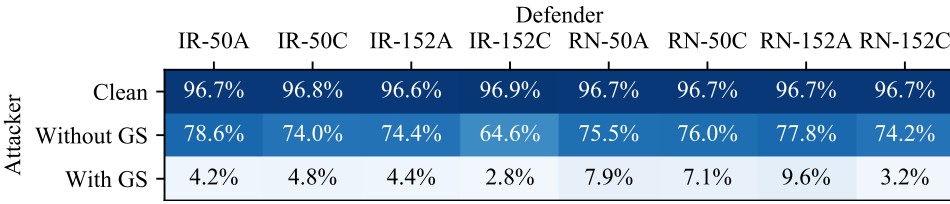

|  | Defender | | | | | | | |
|  | IR-50A | IR-50C | IR-152A | IR-152C | RN-50A | RN-50C | RN-152A | RN-152C |
|---|---|---|---|---|---|---|---|---|
| Clean | 96.7% | 96.8% | 96.6% | 96.9% | 96.7% | 96.7% | 96.7% | 96.7% |
| Without GS | 78.6% | 74.0% | 74.4% | 64.6% | 75.5% | 76.0% | 77.8% | 74.2% |
| With GS | 4.2% | 4.8% | 4.4% | 2.8% | 7.9% | 7.1% | 9.6% | 3.2% |

(The leftmost label "Attacker" runs vertically beside the row labels.)

Table 3: Rank-50 accuracy of FR models tested on blurred LowKey images computed with/without Gaussian smoothing.

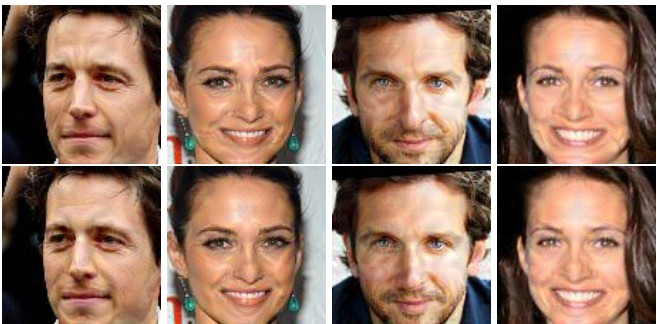

Figure 3: LowKey attacked images computed without (above) and with (below) Gaussian smoothing.

### 6.4 Robustness to Image Compression

Since users may save their images in various formats after passing them through LowKey, the images we produce must provide protection even after being saved in common formats. Our baseline tests are conducted with images saved in uncompressed PNG format. To test performance under compression, we convert protected images to JPEG format and repeat our experiments on commercial APIs. While compression very slightly decreases performance, the attack is still very effective: Microsoft Azure Face is now able to recognize 0.2% of images compared to 0.1% when saved in the PNG format. Likewise, Amazon Rekognition now recognizes 3.8% of probe images compared to 2.4% previously.

### 6.5 Scalability to All Image Sizes (Disclaimer)

Many tools in deep learning require that inputs be of particular dimensions, but user images on social media sites come in all shapes and sizes. Therefore, LowKey must be flexible. Since the detection and alignment pipeline in our attack resizes images in a differentiable fashion, we can attack images of any size and aspect ratio. Additionally, we apply the LPIPS penalty to the entire original image, which prevents box-shaped artifacts from developing on the boundaries of the rectangle containing the face. Since LowKey does not have a fixed attack budget, perturbations may have different magnitudes on different images. Figure 4 shows the variability of LowKey perturbations on very large images; the image of Tom Hanks (first column) is one of the best looking examples of LowKey on large images, while the image of Tina Fey (last column) is one of the worst looking examples. Protecting very large images is a more challenging task than protecting small images because of the black-box detection, alignment, and re-scaling used in APIs which affect large images more significantly. These experiments indicate that users will receive stronger protection if they use LowKey on smaller images.

We test the effectiveness of LowKey on large images by protecting gallery images of 10 identities from Facescrub (with 17 images in the gallery on average) and using 20 probe images per person. We also vary the magnitude of the perturbation to find the smallest perturbation that is sufficient to protect images (Table 4). In this way, we find that users may trade off some protection in exchange for better looking images at their own discretion. Additionally, we find that LowKey works much better with smaller gallery sizes; when only 5 gallery images are used, the performance of Amazon Rekognition drops from 32.5% to 11% in the rank-50 setting. This observation suggests that users can upload new profile pictures less frequently in order to decrease the number of gallery images corresponding to their identity and thus enhance their protection. Finally, the quality of probe images is also important; when small probe images are used, like those which would occur in low resolution security camera footage, the accuracy of Amazon Rekognition drops from 32.5% to 19%.

## 7 Discussion

In this work, we develop a tool for protecting users from unauthorized facial recognition. Our tool adversarially pre-processes user images before they are uploaded to social media. These pre-processed images are useless for third-party organizations who collect them for facial recognition. While we have shown that LowKey is highly effective against commercial black-box APIs, it does not protect users 100% of the time and may be circumvented by specially engineered robust systems. Thus,

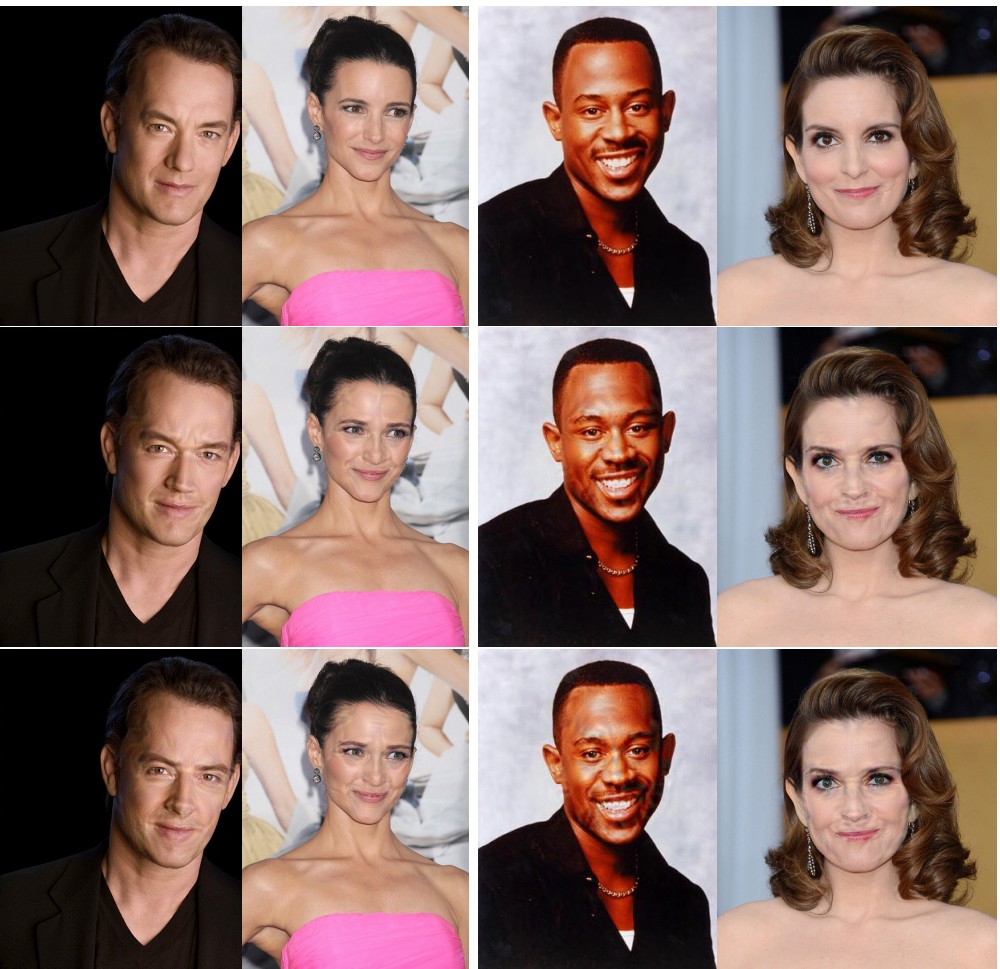

Figure 4: First row: Original large images, Second row: Images protected with LowKey (medium magnitude), Third row: Images protected with LowKey (large magnitude).

|  | Amazon rank-1 | Amazon rank-50 | Microsoft rank-1 |
|---|---|---|---|
| Clean | 89.0% | 98.5% | 86.0% |
| LowKey 10 | 63.0% | 94.5% | 75.5% |
| LowKey 20 | 34.0% | 59.5% | 30.5% |
| LowKey 30 | 20.5% | 36.5% | 12.7% |
| LowKey 40 | 14.5% | 36.0% | 3.0% |
| LowKey 50 | 11.0% | 32.5% | 0.0% |

Table 4: Evaluation of LowKey on full-size images. Rows indicate levels of magnitude of LowKey (denoted by the number of attack steps).

we hope that users will still remain cautious about publicly revealing personal information. One interesting future direction is to produce adversarial filters that are more aesthetically pleasing in order to promote wider use of this tool. However, it may be that there is no free lunch, and one cannot fool state-of-the-art facial recognition systems without visible perturbations. Facial recognition systems are not fragile, and other attacks that have attempted to break them have failed. Finally, we note that one of our goals in making this tool widely available is to promote broader awareness of facial recognition and the ethical issues it raises. Our webtool can be found at `lowkey.umiacs.umd.edu`.

## ACKNOWLEDGMENTS

This work was supported by the DARPA GARD and DARPA QED programs. Further support was provided by the AFOSR MURI program, and the National Science Foundation's DMS division. Computation resources were funded by the Sloan Foundation.

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

## 8 APPENDIX

### 8.1 IMPLEMENTATION DETAILS

We train all of our feature extractors using focal loss (Lin et al., 2017) with a batch size of 512 for 120 epochs. We use an initial learning rate of 0.1 and decrease it by a factor of 10 at epochs 35, 65 and 95. For the optimizer, we use SGD with a momentum of 0.9 and weight decay of 5e-4.

For our adversarial attacks, we use $0.05$ for the perceptual similarity penalty, $\sigma = 3$ and window size 7 for the Gaussian smoothing term. Attacks are computed using signed SGD for 50 epochs with a learning rate of 0.0025.

For face detection and aligning models as well as for training routines, we use the face.evoLVe.PyTorch github repository (Zhao, 2020).

### 8.2 RANK-1 ACCURACY ON FACESCRUB DATA

See Table 5.

|  | Defender | | | | | | | |
|  | IR-50A | IR-50C | IR-152A | IR-152C | RN-50A | RN-50C | RN-152A | RN-152C |
|---|---|---|---|---|---|---|---|---|
| Clean | 95.6% | 96.1% | 96.0% | 96.2% | 95.8% | 95.9% | 95.9% | 96.0% |
| Fawkes | 71.2% | 76.2% | 74.4% | 78.2% | 73.9% | 76.0% | 76.0% | 71.2% |
| IR-50A | 0.0% | 3.5% | 0.5% | 6.4% | 5.0% | 8.2% | 8.7% | 12.4% |
| IR-50C | 0.1% | 0.0% | 0.1% | 9.0% | 3.3% | 3.3% | 3.6% | 5.3% |
| IR-152A | 0.3% | 1.8% | 0.0% | 4.2% | 6.8% | 8.9% | 9.4% | 10.8% |
| IR-152C | 0.1% | 0.1% | 0.1% | 0.0% | 1.0% | 1.4% | 2.2% | 2.9% |
| RN-50A | 3.8% | 7.0% | 6.8% | 8.3% | 0.9% | 2.0% | 2.8% | 4.6% |
| RN-50C | 3.1% | 5.7% | 5.6% | 7.9% | 1.2% | 0.1% | 2.0% | 3.5% |
| RN-152A | 1.5% | 3.1% | 2.7% | 4.9% | 0.3% | 0.5% | 0.0% | 0.4% |
| RN-152C | 1.7% | 2.6% | 3.1% | 3.9% | 0.8% | 0.8% | 0.5% | 0.0% |
| Ensemble | 0.0% | 0.0% | 0.1% | 0.0% | 0.2% | 0.4% | 0.6% | 0.1% |

*Attacker* (row label, left vertical axis)

Table 5: Rank-1 accuracy of the LowKey and Fawkes attacks on the FaceScrub dataset. After the first two rows, each row represents LowKey attacks generated from the same model. Each column represents inference on a single model. The first two letters in the model's name denote the type of backbone: IR or ResNet (RN). The last letter in the model's name indicates the type of head; "A" denotes ArcFace, and "C" denotes CosFace. Smaller numbers, and lighter colors, indicate more successful attacks.

### 8.3 RESULTS ON UMDFACES DATASET

We repeat controlled experiments on the UMDFaces dataset which contains over 367,000 photos of 8,277 identities. For UMDFaces, we also choose 100 identities at random and attack their gallery images while keeping one-tenth of each identity's photos as probe images. Experimental results are reported in Tables 6 and 7. It can be seen that the effectiveness of LowKey attacks on the UMDFaces dataset is slightly lower, which is likely a result of the much smaller gallery.

### 8.4 CAN WE REDUCE THE SIZE OF OUR ATTACK?

In order to make our attacks more aesthetically pleasing, we try to reduce the size of perturbation by increasing the perceptual similarity penalty from 0.05 to 0.08. This attack is depicted in Figure 5 as a "LowKey small attack". Unfortunately, even a small decrease in the perturbation size results in a huge decrease in efficiency of the attack. In the rank-50 setting Amazon Rekognition is able to recognize 17.2% of probe images belonging to users protected with a LowKey small attack. Similarly, Microsoft

| | Defender | | | | | | | |
|---|---|---|---|---|---|---|---|---|
| Attacker | IR-50A | IR-50C | IR-152A | IR-152C | RN-50A | RN-50C | RN-152A | RN-152C |
| Clean | 98.6% | 98.3% | 98.6% | 98.6% | 98.3% | 98.3% | 98.6% | 98.6% |
| IR-50A | 0.3% | 38.9% | 25.9% | 48.6% | 47.7% | 55.7% | 57.1% | 62.2% |
| IR-50C | 13.6% | 0.3% | 16.5% | 17.6% | 37.5% | 39.2% | 51.4% | 46.3% |
| IR-152A | 23.0% | 32.1% | 0.0% | 36.4% | 52.8% | 56.5% | 58.2% | 58.5% |
| IR-152C | 8.8% | 8.8% | 9.1% | 1.1% | 26.7% | 28.1% | 34.9% | 31.8% |
| RN-50A | 51.4% | 56.3% | 47.2% | 53.1% | 0.3% | 30.7% | 38.4% | 43.2% |
| RN-50C | 49.4% | 48.3% | 55.1% | 54.0% | 23.0% | 1.1% | 37.2% | 36.4% |
| RN-152A | 30.4% | 38.1% | 39.8% | 43.8% | 18.8% | 22.2% | 3.4% | 25.9% |
| RN-152C | 26.4% | 33.8% | 35.5% | 37.8% | 17.3% | 18.2% | 16.8% | 3.4% |
| Ensemble | 10.5% | 4.5% | 9.7% | 8.8% | 15.1% | 6.5% | 12.8% | 13.6% |

Table 6: Rank-50 accuracy of LowKey attacks on the UMDFaces dataset. After the first row, each row represents LowKey attacks generated from the same model. Each column represents inference on a single model. The first two letters in the model's name denote the type of backbone: IR or ResNet (RN). The last letter in the model's name indicates the type of head; "A" denotes ArcFace, and "C" denotes CosFace. Smaller numbers, and lighter colors, indicate more successful attacks.

| | Defender | | | | | | | |
|---|---|---|---|---|---|---|---|---|
| Attacker | IR-50A | IR-50C | IR-152A | IR-152C | RN-50A | RN-50C | RN-152A | RN-152C |
| Clean | 96.0% | 97.2% | 96.9% | 96.9% | 96.3% | 96.6% | 97.2% | 96.3% |
| IR-50A | 0.0% | 12.2% | 6.3% | 18.2% | 17.6% | 25.9% | 28.1% | 31.3% |
| IR-50C | 1.7% | 0.0% | 2.6% | 4.3% | 11.6% | 13.9% | 17.3% | 21.0% |
| IR-152A | 4.3% | 12.8% | 0.0% | 14.5% | 22.4% | 25.6% | 29.5% | 30.1% |
| IR-152C | 1.7% | 2.0% | 2.6% | 0.0% | 6.8% | 9.7% | 13.6% | 11.4% |
| RN-50A | 26.4% | 31.0% | 13.9% | 26.4% | 0.0% | 9.4% | 15.1% | 19.0% |
| RN-50C | 14.5% | 21.3% | 20.7% | 25.3% | 6.5% | 0.6% | 12.8% | 14.8% |
| RN-152A | 9.7% | 16.2% | 16.8% | 17.9% | 5.1% | 7.1% | 0.3% | 7.4% |
| RN-152C | 7.4% | 10.2% | 11.9% | 13.1% | 3.1% | 4.5% | 5.4% | 0.9% |
| Ensemble | 2.8% | 1.7% | 2.6% | 3.1% | 5.1% | 2.0% | 3.1% | 4.0% |

Table 7: Rank-1 accuracy of LowKey attack attacks on the UMDFaces dataset. After the first row, each row represents LowKey attacks generated from the same model. Each column represents inference on a single model. The first two letters in the model's name denote the type of backbone: IR or ResNet (RN). The last letter in the model's name indicates the type of head; "A" denotes ArcFace, and "C" denotes CosFace. Smaller numbers, and lighter colors, indicate more successful attacks.

Azure Face recognizes 5.5% of probe images. Results of controlled experiments are reported in Tables 8 and 9.

## 8.5 COMPARISON WITH FAWKES

By comparing a set of images protected with LowKey and Fawkes tools, we can see that both attacks are noticeable, but distort images in different ways. While Fawkes adds conspicuous artifacts on the face (such as mustaches or lines on the nose), LowKey attack mostly changes the textures and adds spots on a person's skin. See Figure 5 for a visual comparison.

| | | Defender | | | | | | |
|---|---|---|---|---|---|---|---|---|
| Attacker | | IR-50A | IR-50C | IR-152A | IR-152C | RN-50A | RN-50C | RN-152A | RN-152C |
| | Clean | 96.8% | 96.8% | 96.7% | 96.8% | 96.8% | 96.8% | 96.7% | 96.7% |
| | IR-50A | 1.0% | 41.1% | 24.7% | 55.2% | 49.7% | 65.0% | 64.2% | 68.8% |
| | IR-50C | 17.4% | 2.6% | 20.4% | 30.2% | 47.0% | 49.0% | 56.2% | 55.6% |
| | IR-152A | 28.8% | 38.6% | 0.3% | 49.8% | 61.2% | 66.6% | 69.8% | 70.7% |
| | IR-152C | 14.2% | 14.0% | 16.2% | 2.7% | 37.0% | 38.6% | 44.4% | 42.9% |
| | RN-50A | 49.3% | 62.3% | 64.1% | 68.0% | 1.5% | 35.9% | 41.8% | 49.3% |
| | RN-50C | 57.4% | 59.9% | 62.1% | 64.4% | 31.1% | 3.6% | 46.8% | 48.6% |
| | RN-152A | 42.9% | 51.3% | 52.3% | 55.0% | 25.6% | 32.9% | 5.6% | 35.0% |
| | RN-152C | 41.8% | 48.9% | 47.9% | 52.7% | 28.2% | 29.2% | 30.4% | 8.5% |
| | Ensemble | 18.0% | 21.8% | 19.4% | 12.2% | 23.1% | 24.3% | 27.0% | 14.1% |

Table 8: Rank-50 accuracy of LowKey small attacks on the UMDFaces dataset. After the first row, each row represents LowKey small attacks generated from the same model. Each column represents inference on a single model. The first two letters in the model's name denote the type of backbone: IR or ResNet (RN). The last letter in the model's name indicates the type of head; "A" denotes ArcFace, and "C" denotes CosFace. Smaller numbers, and lighter colors, indicate more successful attacks.

| | | Defender | | | | | | |
|---|---|---|---|---|---|---|---|---|
| Attacker | | IR-50A | IR-50C | IR-152A | IR-152C | RN-50A | RN-50C | RN-152A | RN-152C |
| | Clean | 95.6% | 96.1% | 96.0% | 96.2% | 95.8% | 95.9% | 95.9% | 96.0% |
| | IR-50A | 0.0% | 6.1% | 2.2% | 10.5% | 9.7% | 14.4% | 16.9% | 19.5% |
| | IR-50C | 1.6% | 0.2% | 2.5% | 5.8% | 9.3% | 11.1% | 14.0% | 14.5% |
| | IR-152A | 2.2% | 5.6% | 0.0% | 9.6% | 14.0% | 17.4% | 19.2% | 20.7% |
| | IR-152C | 1.3% | 1.7% | 2.2% | 0.4% | 5.1% | 7.5% | 9.6% | 9.7% |
| | RN-50A | 8.7% | 14.9% | 14.8% | 17.1% | 0.2% | 7.6% | 7.8% | 12.5% |
| | RN-50C | 9.9% | 14.6% | 15.6% | 17.2% | 5.6% | 0.6% | 8.8% | 11.6% |
| | RN-152A | 9.2% | 13.5% | 12.2% | 14.8% | 5.7% | 7.9% | 0.8% | 8.5% |
| | RN-152C | 6.9% | 11.3% | 11.8% | 12.2% | 4.8% | 5.3% | 6.0% | 1.3% |
| | Ensemble | 2.6% | 3.6% | 3.5% | 2.9% | 4.0% | 4.7% | 5.4% | 3.9% |

Table 9: Rank-1 accuracy of LowKey small attacks on the UMDFaces dataset. After the first row, each row represents LowKey small attacks generated from the same model. Each column represents inference on a single model. The first two letters in the model's name denote the type of backbone: IR or ResNet (RN). The last letter in the model's name indicates the type of head; "A" denotes ArcFace, and "C" denotes CosFace. Smaller numbers, and lighter colors, indicate more successful attacks.

## 8.6 GAUSSIAN SMOOTHING IN LOWKEY

For the parameters of the Gaussian smoothing term in the optimization problem (1), we use 3 for $\sigma$ and 7 for window size. For the defensive Gaussian blur, we use $\sigma = 2$ and no window size.

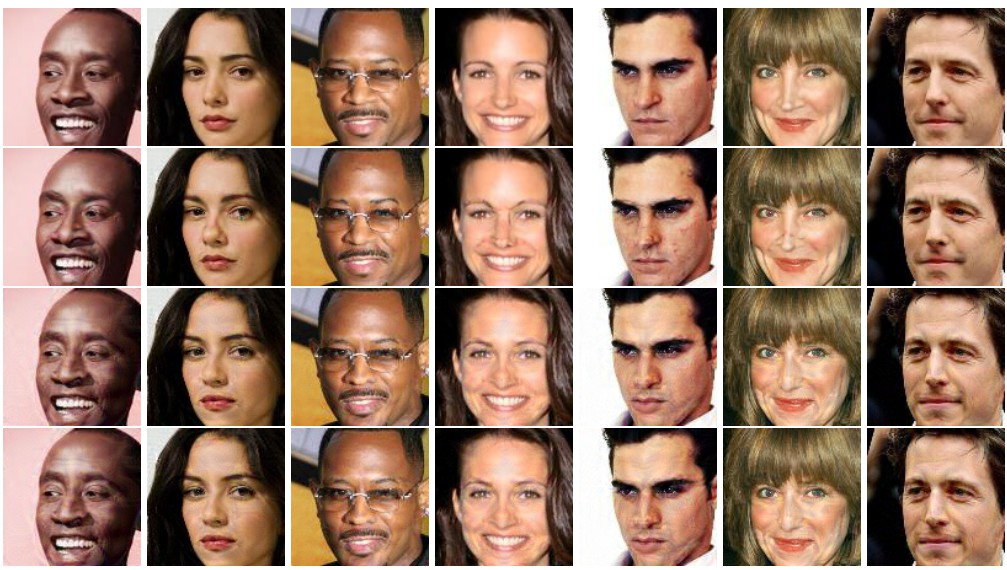

Figure 5: Panel of different attacks. First row: original images, second row: Fawkes attack, third row: LowKey small attack, last row: LowKey attack.

|  | Defender | | | | | | | |
|  | IR-50A | IR-50C | IR-152A | IR-152C | RN-50A | RN-50C | RN-152A | RN-152C |
|---|---|---|---|---|---|---|---|---|
| Clean | 84.4% | 85.3% | 84.7% | 86.1% | 86.7% | 87.9% | 88.5% | 89.6% |
| Without GS | 13.3% | 17.9% | 15.8% | 13.7% | 18.9% | 20.4% | 23.2% | 20.6% |
| With GS | 0.3% | 0.3% | 0.1% | 0.0% | 1.0% | 0.9% | 0.6% | 0.6% |

Attacker

Table 10: Rank-1 accuracy of FR models tested on blurred images attacked without and with the Gaussian smoothing term. The first two letters in the model's name denote the type of backbone: IR or ResNet (RN). The last letter in the model's name indicates the type of head; "A" denotes ArcFace, and "C" denotes CosFace. Smaller numbers, and lighter colors, indicate more successful attacks.

