# OpenReview forum: "LowKey: Leveraging Adversarial Attacks to Protect Social Media Users from Facial Recognition"
_ICLR.cc/2021/Conference — ICLR 2021 Poster_

### Official Review · AnonReviewer2 · 2020-10-21
**Face recognition & User privacy. Great results addressing a hot issue. Fun read.**

**Rating:** 7
**Confidence:** 3

**Review:**

1. Summarize what the paper claims to contribute. Be positive and generous.
The paper claims to contribute the following three:
(1) design a black-box adversarial attack on facial recognition models.
(2) interrogate the performance of the proposed method on commercial black-box APIs, including Amazon Rekognition and Microsoft Azure Face, and against the existing data poisoning alternative, Fawkes.
(3) release an easy-to-use webtool, LowKey.
The result tables on the paper look great!

2. List strong and weak points of the paper. Be as comprehensive as possible.
(1) Strengths
  a. The margins of performance differences between LowKey and Fawkes looks great (74%+ absolute).
  b. The robustness of the model to varying image compression and image size.
  c. Run-time latency improvement of the model against Fawkes.
  d. Thorough analysis involving ablation tests (e.g. effect of Gaussian Smoothing and different expert models).
  e. Including eval metrics of top-50 accuracy.
(2) Weaknesses
  a. Experimental setup descriptions are difficult to follow at times.

3. Clearly state your recommendation (accept or reject) with one or two key reasons for this choice.
Accept because the results look promising even with the model's performance analyzed exhaustively.

4. Provide supporting arguments for your recommendation.
The paper proposes a new benchmark with close-to-practical-performance results in latency, accuracy, and robustness to compression. Digital user privacy is a hot topic to be further studied especially in the context of practical application purposes. And, in that perspective, this paper advances the applied research in the area considerably.

5. Ask questions you would like answered by the authors to help you clarify your understanding of the paper and provide the additional evidence you need to be confident in your assessment.
(1) Curious to learn if there is any good literature that backs the number 50 in *top-50 accuracy*. Is this number arbitrarily chosen to be something greater than 1?
(2) The experiment setup descriptions are a bit difficult to follow.
  a. Based on my understanding, in the experiment, the gallery images can contain "attacked" images but not the probe images.
  b. And, for each identity in the dataset, either all or none of the identity's gallery images are "attacked".
  c. None of the 1 million distractor images are "attacked".
  d. All of the 1 million are used as gallery images.
Did I get the experiment set-up right?
(3) Follow-up question to (3), in the real-world, wouldn't be messier with the gallery images containing both "attacked" and clean images of the same identity? The same for probe images. How will LowKey perform in this messy real world?
(4) Follow-up question to (3) and (4), how will LowKey perform if the same identity's image and probe images are "attacked"?
(5) How are the IR-* and RN-* models ensambled? Averaging? Majority voting? Something else? This helps with reproducibility.
(6) In the section 6.3 *Run-time*, the paper says *While Fawkes averages 54 seconds per image, LowKey only averages 32 seconds per image*. Is this an end-to-end latency including blurring and face detection?
(7) What's the URL to use LowKey service?

6. Provide additional feedback with the aim to improve the paper. Make it clear that these points are here to help, and not necessarily part of your decision assessment.
First, Great work! I enjoyed reading your paper. Looking forward to trying LowKey.
Have you tried measuring the performance of LowKey on [OpenFace](https://cmusatyalab.github.io/openface/) and [Kairos](https://www.kairos.com/)?

---

> ### Author Response · Authors · 2020-11-24
> **Thank you for the review**
>
> Thank you for your detailed feedback.  Regarding your questions:
>
> 1. Top-50 accuracy is arbitrary.  We have not seen a standardized number for works that evaluate on higher than rank-1 accuracy, but we have been advised that law enforcement (as well as commercial APIs in our own experience) may produce lists of potential candidates rather than a single match, and this seems like a regime worth testing.
>
> 2. Your understanding of our experimental setup is absolutely correct.  We have updated our draft to explain our experimental setup in closer detail.
>
> 3. We agree that some users may have images on the internet that are out of their control, and this may pose a problem for those users.  Unfortunately, if an organization already has a large database of clean images corresponding to an individual, additional adversarial examples will not fool facial recognition systems.
>
> 4. In our work, we assume that users do not have control over probe images (because probe images come from other sources such as security camera footage acquired in a building lobby or a traffic intersection, and are often obtained without the target person even knowing). LowKey protects the gallery images so that they cannot be used to recognize a user on a probe image taken by a third party.
>
> 5. In order to ensemblize models, we compute the normalized distance to the clean image in the feature space of each model, and we average over all models. The perturbation is then computed using gradient ascent on the average distance in the feature space of ensemble models.
>
> 6. Regarding our runtime comparison, the numbers indeed reflect an end-to-end latency including blurring and face detection.
>
> 7. In order to preserve anonymity, we will update our paper to include the URL after ICLR decisions are released.  The tool is currently hosted under our institution’s web domain, and we cannot figure out a way to make this available without divulging our institutional affiliations.

---

### Official Review · AnonReviewer3 · 2020-10-26
**An adversarial tool against face recognition in social media**

**Rating:** 6
**Confidence:** 4

**Review:**

This paper presents a method/tool, i.e., LowKey, to protect user privacy which leverages adversarial attacks to pre-process facial images against the black-box facial recognition system in social media, yet the processed facial images remain visually acceptable. The LowKey method proposes to attack an ensemble of facerec models by optimizing the Gaussian blur to the original face images, with the LIIPS metric on the L2 distance in the \emph{feature space}. Thus, the processed face images remain visually legible to human. The ensemble of facerec models include ResNet-50, RestNet-152, IR-50 and IR-152 trained on MS-Celeb-1M dataset. The LowKey method has demonstrated very effective in combating black-box commercial facerec system at Amazon and Microsoft.

The proposed LowKey method to attack facial recognition is social media is an interesting work. The technical approach appears sound, yet some technical details may need more discussion. The experiments are thorough and convincing. The experiments on commercial facial facerec system on Amazon and Microsoft are very interesting.

Some detailed comments:

1）	In Eq.1, is there a single Gaussian blur $G$ on the whole facial image or the whole image? The optimization on $x^{‘}$ is equivalent on optimizing the parameters of the Gaussian blur kernel of $G$, right? In Sec.8.1, the Gaussian kernel seems a fixed one, then the optimization of processed image is on the location of the kernel? Please elaborate more on the training for Eq.1, perhaps move Sec.8.1 below Eq.1.

2）	How to handle multiple faces in an image. The image detector A seems in the loop of Eq.1. If the faces are hard to detect to the facial recognition system, this may be an even better way to protect user privacy, because the facerec model may evolve by online learning in the commercial system, yet the face detectors are pretty much fixed.

3）	Any experiment and comments on attacking the large social media website of Facebook? A relevant reference: Deep-Face: Closing the Gap to Human-Level Performance in Face Verification. CVPR 2014.

Overall, this is an interesting work addressing the user privacy protection against commercial facial recognition system. The technical approach is sound, yet needs more explanation and discussion.

---

> ### Author Response · Authors · 2020-11-24
> **Thank you for your review**
>
> Thank you for your interest and for your feedback.  Regarding your comments:
>
> 1.  In Eq. 1, G is a Gaussian blur of the whole image, and it has fixed radius and window size (so we do not optimize parameters of G). Intuitively, Gaussian blur helps to protect an image even if it is smoothed by a third-party before face detection and recognition (for example, during re-sizing/interpolation to a standardized resolution, or as a simple defense to remove adversarial “noise”). In particular, LowKey moves both “protected’’ and “protected+blurred’’ versions of an image away from the original image in feature space.  Thank you for bringing up this ambiguity, and we explain Gaussian blur in more detail in the updated version of our paper.
>
> 2. There are works regarding adversarial attacks on object detection.  However, the drawback of this approach is that it is easily detectable and face extraction can be performed manually by simply drawing a bounding box around the face, thus nullifying the attack.  Perhaps, it is possible to combine LowKey with an objectness score loss so that the attack both confuses facial recognition feature extractors and flies under the radar of face detection.
> (references: Adversarial Attacks on Face Detectors using Neural Net based Constrained Optimization 2018, Making an Invisibility Cloak: Real World Adversarial Attacks on Object Detectors 2019)
>
> 3. Thank you for pointing out this work.  Testing LowKey against Facebook’s facial recognition system would be interesting.  To our knowledge, this work uses a proprietary dataset of face images from Facebook that still is not available to the public.  Moreover, we are not sure how to query Facebook’s facial recognition system in a controlled, systematic, and legal manner for testing.  We will look into this possibility.

---

### Official Review · AnonReviewer1 · 2020-10-29
**Review of the manuscript LowKey: Leveraging Adversarial Attacks to Protect Social Media Users from Facial Recognition**

**Rating:** 7
**Confidence:** 5

**Review:**

*********
Summary Of The Manuscript:
*********
This manuscript focuses on the problem to protect the users/humans from unauthorized facial recognition systems. To tackle the issue mentioned, the author proposes a customized adversarial filter to work against industrial/government facial recognition systems. In addition, the author claims that their easy-to-use web tool helps to significantly degrade the accuracy below 1% of AWS Service - Amazon Rekognition and Microsoft Azure Face Recognition API for face recognition and similar systems.

*********
Strength Of The Manuscript:
*********
Clarity:
++ The paper reads very well and provides a very good description of related work and background, motivating the problem. Even outside of the contribution of this paper, I would recommend this paper to people getting started with protecting users' information from industrial systems as it provides a thorough description of the part of the pipelines it deals with.

Novelty:
There are mainly two points are novel which are presented very well and are as follows:
++ The author designs a custom black-box adversarial attack on facial recognition models where the proposed algorithm changes the representation of the features in such a way that it will preserve the image quality.
++ The in-depth analysis with the commercial - Amazon Rekognition and Microsoft Azure Face Recognition APIs shows the practical usage of the author's proposed adversarial attack.

Experiments:
++ There are a number of experiments performed across datasets that are extensive and fair. The fact that the proposed adversarial attack achieves better results while preserving the image quality, makes me confident in the result as the implementation and experiments are sufficient and presented in a good manner. Additionally, the improvements are fairly consistent. Besides, in-depth analysis/ablation studies are done on the robustness of the approach, and comparison with the commercial APIs provides a thorough analysis of where the benefits of the author's approach are obtained.

Reproducibility:
++ First of all, I would like to thank the author for providing code in the supplementary material and because from the code, most of my doubts have been solved and I can say this that the code is written in a very good manner and helped me to understand the whole pipeline of their work and I appreciate the authors for their effort.

*********
Weakness Of The Manuscript:
*********
Overall, apart from the contribution of the approach, I have some concerns regarding the manuscript.

-- Can the authors provide a brief description of the core difference between their proposed adversarial attack (LowKey) and attack generated by any good GANs based model (i.e. Style GAN.). I believe that the standalone contribution of the manuscript is to create such an attack to hide the user's identity. Thus what makes a user believe that the proposed webtool will be helpful to hide their identity if it only applies a LowKey adversarial attack.

*********
Justification Of The Review:
*********
-- In the reviewer's opinion, in its current form, the paper provides in-depth analysis for protecting the user's identity from any industrial/government surveillance facial recognition system and the authors did an excellent job to provide a brief insight on their complete pipeline. I appreciate the author's efforts to provide a code in supplementary material which has nullified my doubts in such a manner that makes me believe that the work is incremental and should reach the Computer Vision community. Therefore the current rating of the paper will be 7 in reviewers' opinion because of fairly consistent work and practical usage.

---

> ### Author Response · Authors · 2020-11-24
> **Thank you for the review**
>
> Thank you for your interest and for your comments.
>
> Adopting a GAN-based attack seems like a promising direction for accelerating LowKey.  The primary reason we avoided this approach is that GAN-based adversarial attacks have not been very successful on high-dimensional data.  Adversarial attack GANs generally operate on low-dimensional data like MNIST and CIFAR-10 (Generating Adversarial Examples with Adversarial Networks 2018, AI-GAN: Attack-Inspired Generation of Adversarial Examples 2019).  Still, we agree that GANs are an interesting future direction. Such a method would require a GAN whose input is an image rather than a random latent vector.  StyleGAN does not have this property, but with some engineering, these difficulties could likely be overcome.

---

### Comment · ~Yinpeng_Dong3 · 2021-02-24
**A closely related work**

Dear authors,

Congratulations on the acceptance of this work. This is a good paper that leverages adversarial attacks for privacy protection.

I want to point out a closely related work done by our group:

"Towards Privacy Protection by Generating Adversarial Identity Masks", arXiv preprint 2003.06814.

Our work also aims to protect user privacy against face recognition systems by applying adversarial perturbations, although the techniques are different from your work in some aspects. We appreciate it if you could discuss with our work in your paper.

Thanks,
Yinpeng

---

> ### Author Response · Authors · 2021-02-24
> **Thanks for bringing your work to our attention!**
>
> Hi Yinpeng,
>
> Thanks for reaching out.  We apologize for missing your work in our initial literature review.  We will take a closer look at your work and consider discussing it in our next draft.
>
> -Micah

---

### Decision · Program_Chairs · 2021-01-07
**Final Decision**

**Decision:**

Accept (Poster)

**Comment:**

This paper presents a method named LowKey, which is designed to protect user privacy. This is done by taking advantage of adversarial attacks to pre-process facial images against the black-box facial recognition system in social media, yet the processed facial images remain visually acceptable. The paper experimentally illustrates that it is effective against two existing commercial facial recognition APIs.

The reviewers unanimously agree that this is an interesting and important problem, and recommend the paper for acceptance. The ACs agree.